# Cancer-Related Symptom Management Intervention for Southwest American Indians

**DOI:** 10.3390/cancers14194771

**Published:** 2022-09-29

**Authors:** Felicia S. Hodge, Tracy Line-Itty, Rachel H. A. Arbing

**Affiliations:** 1School of Nursing, University of California, Los Angeles, 700 Tiverton Avenue, Room 5-934A Factor Building, Los Angeles, CA 90095, USA; 2Fielding School of Public Health, University of California, Los Angeles, 640 Charles E Young Drive South, Los Angeles, CA 90095, USA

**Keywords:** American Indian, cancer, survivors, symptom management, quality of life, intervention

## Abstract

**Simple Summary:**

Quality of life during, and even after, cancer treatment is greatly affected by cancer symptoms that include pain, fatigue, and changes to mental state and activities of daily living, to name a few. American Indians living in the Southwestern United States have cancer experiences which may be different than the general population and have long been understudied. A randomized controlled trial designed to test the impact of a culturally tailored intervention on the management of individual cancer symptoms was implemented. Outcomes included improvement in pain, depression, fatigue and loss of function management in adult American Indians. Study evaluations at post-test show a significant improvement in scores from pre-test and compared to the control group, demonstrating increased knowledge levels in managing cancer-related symptoms. Study findings guide researchers towards a better understanding of the meaning and impact of cancer symptoms for American Indian cancer survivors, thus their improving care and quality of life.

**Abstract:**

There is limited literature related to culturally embedded meanings of cancer and related symptoms among American Indians. A culturally appropriate intervention to improve management of cancer-related symptoms, including pain, depression, fatigue and loss of function, was tested. Two-hundred and twenty-two adult American Indians with cancer were recruited from eight Southwest sites for a randomized clinical trial. The intervention group received tailored education, a toolkit with a video, and participated in discussion sessions on cancer symptom management; the control group received information on dental care. Pre- and post-test questionnaires were administered to control and intervention groups. Measures included socio-demographics, cancer-related symptom management knowledge and behavior, and quality of life measures. Male cancer survivors reported poorer self-assessed health status and lower scores on quality-of-life indicators as compared to female cancer survivors. Significant improvement was reported in symptom management knowledge scores following the intervention: management of pain (*p* = 0.003), depression (*p* = 0.004), fatigue (*p* = 0.0001), and loss of function (*p* = 0.0001). This study is one of the first to demonstrate a change in physical symptom self-management skills, suggesting culturally appropriate education and interventions can successfully enhance cancer-related symptom management knowledge and practice.

## 1. Introduction

Cancer is a chronic illness that places additional demands on cancer survivors and their families. American Indians and Alaska Natives are at higher risk for some cancers than the general population. The Centers for Disease Control and Prevention (CDC) reports that American Indians and Alaska Natives are more likely to be diagnosed with, and have higher rates of, certain cancers such as lung, colorectal, liver, stomach, and kidney cancers, than non-Hispanic Whites [1]. Following a cancer diagnosis, treatment can include surgery, chemotherapy and radiation, and American Indian survivors face many additional challenges in managing their healthcare. In addition to long-term surveillance and possible additional treatment, survivors contend with serious cancer-related symptoms, including pain, depression, fatigue, and loss of function. Having adequate skills to respond to cancer symptoms and implement self-management strategies is an essential part of cancer survivorship.

The role of self-management of cancer-related symptoms is broader than simply responding to the physical problems experienced after cancer treatment. Shifting personal perspectives from illness to wellness reinforces holistic cancer care management. Problem solving includes the ability to identify the source of a problem and resources needed, and then acting on the steps needed to improve daily living and thereby quality of life. Maintaining regular medical appointments and on-going surveillance of pain, fatigue and loss of function takes organizational skills and follow-through. Recognizing symptoms and seeking appropriate care is necessary for healthy well-being. Having knowledge of strategies to improve daily living and communication skills is helpful in the self-management process.

Several studies point to improved quality of life [2,3], health [2,4], and psychological and emotional well-being [5,6] as essential goals in the self-management process among cancer survivors. It is also critical that strategies to promote health and wellness are sensitive to survivors’ beliefs and respect cultural traditions. Burnette, Roh, Liddell, and Lee [4] conducted a qualitative study in South Dakota identifying American Indian women cancer survivors’ needs and preferences, with a particular emphasis on community supports for their cancer experience. Participants identified a need for more community-based support systems and infrastructures to ameliorate the cancer survivor experience. The need for an improved healthcare system that included the integration of spirituality and holistic healing options was emphasized. Recommendations for a community approach to raise awareness, education, and support for American Indian cancer survivors was provided.

Education and supportive interventions by healthcare providers, as well as family and caregivers, can also improve the skills and confidence needed to manage cancer-related symptoms [7,8,9]. Educational information can be readily obtained at various clinics and non-profit agencies, such as the American Cancer Society. The Indian Health Service also provides pamphlets and videos on cancer screening and cancer care. Encouragement and practical support through needed transportation and daily living tasks can be instrumental in survivors’ ability to accept and adopt the needed skills to manage cancer-related symptoms.

Despite cancer rates being twice as high for liver cancer (18.1 vs. 7.1/100,000) and near double for stomach and kidney cancers in American Indians compared to non-Hispanic Whites [10], there has been limited scholarly literature, as well as understanding, related to culturally embedded meanings of cancer and related symptoms among American Indians [11,12]. By listening to individual and family cancer experiences and building on everyday cultural values and strengths of a community, effective cancer-related interventions that are relevant and culturally appropriate may be developed.

This paper reports on a randomized control trial (RCT) designed and coordinated to test a culturally sensitive intervention targeting knowledge and strategies to help American Indian cancer survivors and their families/caregivers to better manage cancer-related symptoms. The aim of the study was to explore the cancer experience and barriers to the management of pain, depression, fatigue and loss of function among American Indians residing in the Southwestern United States. Native language and cultural differences in project educational materials and skill building curriculum, as well as in the study instruments, was respected and incorporated. Improving communication about treatment with healthcare providers, such as asking focused follow-up questions and recording easily forgotten information and instructions at medical visits, was one of many key study intervention strategies emphasized in this research intervention curriculum. This study postulated that good communication between patient and provider would facilitate improved cancer knowledge and improved treatment compliance.

The study was carried out over a seven-year period, from project planning, design and coordination to implementation and evaluation. The study was organized into the following three phases: 1. Interviews, 2. Focus Groups, and 3. Intervention Testing. This paper reports on findings and implications from the intervention (Phase 3).

## 2. Materials and Methods

Interested tribal councils and health clinics provided written approvals for research in their communities. Institutional Review Board (IRB) approvals were obtained at the beginning of the study from the University of California, Los Angeles and from the Phoenix Area Indian Health Service. Figure 1 illustrates the study design.

### 2.1. Participant Recruitment

Recruitment efforts consisted of community flyers and notices placed in Tribal/Indian Health Service clinics, and word-of-mouth recruitment lasted two months. The flyers and notices included a description of the project and the contact telephone number and e-mail for those who had questions or who wished to enroll in the study. Participant eligibility criteria (American Indian, age 18 and older, diagnosed with cancer by a medical provider, and resident of the state of Arizona) and the location where the scheduled educational sessions would be held was included. Identified key staff at the clinics/hospitals assisted in distributing the announcements and signing up the interested participants. Project staff made regular visits to sites to register eligible participants into the intervention phase and to read and administer the active consent forms. Three hundred individuals responded to the recruitment efforts and 222 met the study criteria and participated in the intervention phase of the study. Assignment to the Intervention and Control groups were by random assignment using a computerized random assignment program.

The project’s educational “toolkit” that served as the study intervention was comprised of culturally sensitive materials. The Toolkit titled, “Weaving Balance Into Life,” incorporated American Indian values of health and balance into self-management of the most common and debilitating cancer symptoms [13]. In addition to increasing knowledge about cancer itself and productive strategies for relieving symptoms, educational targets also included approaches for building support and reducing communication barriers among caregivers and “Western” healthcare providers. The tools also included resource materials and culturally appropriate instruments for measuring symptoms that intended to improve American Indian knowledge of, and access to, local cancer symptom management services.

Toolkit development incorporated important themes and findings gleaned from qualitative analysis of interview and focus group transcripts from prior phases of this study. The intervention toolkit, developed to be informative and culturally appropriate, used Southwest Native imagery and themes, including American Indian healing practices and spirituality. The toolkit components included a Cancer Symptom Management educational video, a Cancer Symptom Management Guide (six chapters, skills building exercises, and a glossary), a Cancer Resource Directory (contact information for resources by region), Talking Circle Curriculum Slides and “Fact Sheet” review handouts, along with a journal, pen, post-it pad, and a hand-held back massager. The Cancer Symptom Management video reinforced American Indian survivors’ stories about their cancer diagnosis experience, management of cancer-related symptoms, and recommendations to others on health and wellness. American Indian survivors participated in the storytelling phase of the video—cultural advisors from Southwestern tribal groups reviewed and enhanced cultural appropriateness of the study materials. Emphasis for curriculum development was drawn from the needs and desires of the target audience, with goals to improve management of individual symptoms, as well as increase survivor advocacy in healthcare settings [14].

The Intervention was tested at eight locations: four urban sites and four reservation locations. Study participants were randomly assigned to intervention or control groups with the pre- and post-test questionnaires gathered 8 weeks apart.

### 2.2. Intervention Group 

The intervention group received the educational curriculum, cancer information and instruction on how to self-manage cancer symptoms. Participants met weekly at a series of one to one and one-half hour meetings comprised of 15–20 members who met for two months (total of 8 sessions). The meetings were held in a “Talking Circle” format where participants and the facilitator sat in a circle and participants took turns responding to the weekly topic. Seated in a circle, no particular individual is at the head leading the topic, thus all members have equal weight in the discussion of cancer-related matters. At the first session, the trained American Indian facilitator introduced the project and each participant read and signed a consent agreement and completed the 60-minute pre-test questionnaire. Educational materials and a new toolkit chapter were delivered at the subsequent six intervention Talking Circle sessions. The post-test questionnaire was administered at the 8th session. The intervention facilitators discussed with participants a brief lesson following a curriculum guide. Participants then took part in discussions to share and discuss the curriculum information. Participants were encouraged not to use their real names and asked not to discuss personal details about other participants outside of the sessions in order to protect their confidentiality. Each week participants received a new educational component to build their toolkit. All Talking Circles were audiotaped to ensure that the lessons were being taught in a standard manner and to capture important themes that were discussed. A research moderator and an assistant monitored the tape recorder while taking any necessary notes. All moderators received training in focus-group implementation in American Indian populations. Refreshments were offered to participants, as is the custom at American Indian gatherings. Participants received a gift card for travel and other costs associated with participating in the sessions. At the last session, the post-test questionnaire was administered (session 8) and participants received a certificate of completion.

### 2.3. Control Group

The control groups met for the pre-test questionnaire and received information on dental care at their initial visit. The control groups’ post-test questionnaire was administered at week 8. At the end of the project, all participants in the control arm received all toolkit materials.

### 2.4. Measures

#### 2.4.1. Demographic Characteristics

Measures included age, gender, tribal affiliation, degree of Indian blood (reported as 25%, 50%, 75%, or 100%), language (English, Spanish, or tribal language), marital status, number of household members (children and adults), and educational attainment (high school degree and above vs. fewer years of education).

#### 2.4.2. Cancer History

Participants were asked if they have ever been told by their healthcare provider that they have cancer, the type of cancer (e.g., sarcoma or carcinoma), location of where the cancer was found (e.g., breast, colon, etc.), if they were being treated (currently or in the past) and type of treatment (chemotherapy, radiation, surgery, traditional method, other).

#### 2.4.3. Cancer Symptoms

Participants were asked if they experienced pain, depression, fatigue, or loss of function due to their cancer. They were asked to describe the experience (type of pain, depression symptoms, fatigue and limitations in activities of daily living due to their cancer such as mobility, work, social events, and self-care). They were asked what medicines they took for the treatment of symptoms and what they felt worked. They were also asked if they talked to their healthcare providers and/or their family about their symptoms.

#### 2.4.4. Knowledge and Behaviors

Knowledge level of skills needed to manage cancer-related symptoms was measured via a series of true and false questions. Participants were asked if they felt they had or acquired the skills needed to manage cancer-related symptoms.

#### 2.4.5. Quality of Life

Participants were asked about their daily life activities and ability to function, and the impact cancer and cancer treatment had on their lives. Five domains were measured: mobility, self-care, ability to perform usual activities, pain or discomfort, and depression.

## 3. Results and Discussion

Table 1 reports on the participant baseline characteristics. Two hundred and twenty-two participants enrolled in the study. The study sample was heavily skewed toward females with only 30% representing males—a common occurrence in cancer survivorship studies with Indigenous peoples [11]. The mean age was 43 years. About sixty-three percent of participants had less than a high school education, and the majority were unemployed and not in a relationship.

### 3.1. Cancer Diagnosis and Treatment

The most common type of cancer diagnosed among males was prostate (10.0%), followed by colon/rectal cancer (7.1%), then stomach (2.9%) and lung cancer (2.9). Females reported diagnoses of breast cancer (10.6%), followed by ovarian (2.5%), colorectal (1.9%) and kidney cancer (1.9%). Forty-two percent of males and 53.7% females were currently being treated for their cancer. Thirty percent of participants reported that they were treated with chemotherapy for their cancer and 92% were being treated for carcinomas (Table 1). Although 75.0% of males and 62.3% of females received treatment for cancer in the past, the majority of participants (55.6%) were in the mist of treatment for their cancer; 40% of males reported having had surgery and 41% of females reported radiotherapy as the primary treatment they were undergoing for their cancer at the time of the questionnaire. Only about 3% of the study population reported use of traditional methods (such as healing ceremonies, herbal medicines, and traditional diets) for treatment of their cancers, though which methods were used by participants was not characterized in this study. It is noteworthy that, increasingly, Indigenous cancer survivors have derived perceived cultural, spiritual, and emotional benefits from its use in coping and healing from cancer [15,16].

### 3.2. Health Status/Quality of Life at Baseline

Information on the health status and quality of life of cancer survivors is important in that it provides useful information about their ability to function in areas of mobility, self-care, daily activities, and during episodes of pain and depression. Pre-test results demonstrate that male cancer survivors report poorer self-assessed health status and lower scores on quality-of-life indicators as compared to female cancer survivors. For instance, more males (31.6%) than females (24.3%) reported poor physical health interfered with their normal social activities with family, friends, neighbors or groups “quite a bit” (during the past 4 weeks). In addition, more males than females (47.4% vs. 16.2%) reported mobility problems and “difficulty performing work or activity,” and “accomplished less than you would like” (47.4% male and 40.5% female). Although more females than males reported that they suffered from cancer-related pain (2/3 of females and half of males), in the area of emotional health (depression), male survivors reported their emotional health was currently “much worse” (10.5%), which was almost four times that reported by female survivors (2.7%). Loss of function experienced may have contributed to male survivors’ lower scores in emotional health and poor physical health that interfered with normal social activities. Further, it must be acknowledged those who have a holistic view of health and wellness that encompasses a balance of one’s physical, spiritual, emotional, as well as mental well-being may perceive their health differently than those who place more emphasis on physical health alone.

### 3.3. Cancer Symptoms

#### 3.3.1. Pain

Pain is a prevalent symptom in cancer survivors and impacts thinking, concentration, and activities of daily living [17]. In a meta-analysis of 122 studies reporting cancer pain prevalence in adults, rates of pain were 39.3% after curative treatment; 55.0% during anticancer treatment; and 66.4% in advanced, metastatic, or terminal disease; additionally, moderate to severe pain was reported by 38% of all patients [17], indicating a need for improved pain management. This study found that the majority of cancer survivors (67.5% females vs. 49.2% males) suffer from pain due to their cancer experience. The onset of painful episodes occurred before, during, and even after remission, either as a result of the cancer itself or due to the cancer treatment. Although the majority of survivors were prescribed pain medication by their healthcare providers (71.5% females vs. 59.7% males), many survivors did not take pain medication as prescribed due adverse side effects of nausea/drowsiness and fear of becoming addicted to the medication (44.4% females and 39.7% males). Cancer survivors reported that they were generally instructed by their healthcare provider on how to manage or control their pain and that their providers reportedly addressed concerns about the side effects of the pain medication. However, overall, only 49.1% (54.4% females and 37.1% males) had ever been told how to manage pain at time of pre-test. Further, less than one-half (47.5% vs. 42.9% males) reported that they actually knew how to manage their pain. Additionally, at pre-test, 85% females and 68.6% males reported they would like to learn how to manage pain. At pre-test, more than one-half of survivors felt that they would not be able to control (manage) their pain (58.6%). This improved to 29% at post-test (*p* = 0.09). In addition, 67.9% of survivors reported at post-test that they now knew how to manage their pain, a significant increase as compared to the control group (*p* = 0.003). In addition, there was a reduction at post-test of those who reported, “there will always be pain with cancer,” as compared to the control group (*p* = 0.002).

#### 3.3.2. Depression

Depression is a common symptom of cancer with pooled prevalence estimates of 8–48% survivors affected that differ based on cancer treatment phase, type of cancer and/or location of tumor, and type of instrument used [18]. The risk for depression far exceeds that found in the general population, with odds being five times higher in cancer survivors [19]. Since it may resemble neurovegetative symptoms, including sleep disturbance, fatigue, and loss of appetite, a depression diagnosis may often be overlooked, especially in the context of busy oncology units where clinicians are not often skilled at diagnosing mental illness and survivors are reluctant to talk about their emotional health [20]. Depression may extend far beyond cancer treatment [18,21,22], making self-management skills even more important for improving quality of life during survivorship. Many participants in this study were hesitant to discuss having depression themselves, although they felt more comfortable calling it “the blues.” Acknowledging that depression can exist during various times during cancer treatment, as well as during post-treatment, and understanding that depression is treatable is important for survivors and their families. At pre-test, the majority of participants (69% females and 66.2% males) reported that they feel depressed or “get the blues” “now and then.” Sixteen percent of females and 11.6% of males were currently being treated for depression in the form of medication (14.4% females and 10% males) or counseling (6.9% females and 57% males). However, less than half of the survivors (42.5% females and 38.6% males) had been told how to manage their depression or how to manage their life around their medication’s side effects (76.7% males and 57.3% females). As a result, 20.5% had concerns about the side effects of medication used to treat depression. A large percentage of participants (78%) reported at pre-test that they would like to learn how to manage their depression (88.2% females and 68% males). Following the intervention, post-test results showed that a significant increase was reported among participants who now knew how to manage their depression (*p* = 0.004), as compared to the control group. This was an increase in knowledge levels from 36.1% at pre-test to 66.4% at post-test.

#### 3.3.3. Fatigue

Feeling fatigue means being so tired that it interferes with daily activities. Cancer-related fatigue is a very common symptom of cancer before, during, and following cancer treatment and it can affect survivors for long periods of time, yet it can go underrecognized [23]. A meta-analysis of 129 studies dating back to the year 1993 estimated prevalence of fatigue to be 49% in patients with cancer, with major differences related to type of cancer, cancer stage, and gender [24]. Understanding that fatigue is a cancer symptom which can be managed is an important message for the survivor, as well as family members and caregivers who often have to adjust their roles in response. Gaining a better understanding that fatigue is a legitimate cancer and treatment symptom and gaining strategies to manage fatigue were learning goals for the project’s participants. At pre-test, a majority of participants reported experiencing some fatigue. Less than half of survivors (38.8% females and 30% males) reported knowing how to manage their fatigue and more females than males (41.9% vs. 27.4%) reported that they had been told how to manage fatigue time of pre-test. A large majority (68.1% females and 60% males) reported ever having thought about managing fatigue, and an even larger majority (86.3% females and 68.6% males) reported they would like to learn how to manage their fatigue. At post-test, a statistically significant increase was reported among those participants who now knew how to manage their fatigue (67.2% post-test vs. 32.5% pre-test; *p* = 0.0001). The control group reported no change in knowing how to manage their fatigue.

#### 3.3.4. Loss of Function

Many cancer survivors report loss of physical functions due to the cancer itself and/or due its treatment, with older survivors experiencing greater losses [25]. Loss of function (in all or part of the body) is a common symptom that cancer survivors face and substantially impacts their quality of life. Functional limitations may come about shortly after treatment initiation and resolve at its completion, but others may last for years [26]. Irrespective of onset, functional limitations may affect one or more systems, including cardiovascular, pulmonary, and musculoskeletal, and may include additional symptoms, such as peripheral neuropathy, pain, fatigue, and sleep disturbances [26]. Common musculoskeletal limitations include the ability to walk (partial or full loss), weakness in arms and legs, and inability to easily lift articles, as well as sensory limitations, such as in hearing and sight, and cognition loss in area of memory. There is a known link between functional decline and caregiver dependency, impaired quality of life, comorbidity burden and increased mortality [25]. Early detection and use of evidence-based interventions may partly mitigate risk of functional decline in cancer survivors [25]. Learning how to manage loss of function during daily activities was a learning goal of this project. At pre-test, one-third of participants reported that they knew how to manage loss of function (31.3% females and 32.8% males). In addition, about a quarter of males (28.6% vs. 40% females) had been told how to manage loss of function due to cancer. A majority of participants reported that they had ever thought about managing loss of function, and at pre-test 87.5% females and 71.4% males reported that they would like to learn how to manage loss of function. At post-test, a statistically significant increase was reported among those participants who now knew how to manage their loss of function (61.9% post-test vs. 28.5% pre-test; *p* = 0.0001). The control group reported no change in functional status.

### 3.4. Knowledge and Symptom Control

Low pre-test scores largely improved at post-test in the intervention group and comparisons with the control group scores showed significant improvement in scores in all targeted categories. This was observed in the survivors’ level of knowledge and perceived ability to control their cancer-related pain, knowledge of depression symptoms, and perceived ability to adopt recommended skills in managing depression, fatigue and function. Participant scores improved greatly at post-test among participants who reported that they now knew how to manage their cancer-related symptoms. Statistically significant improvement in scores when intervention groups were compared to the control groups was found in the areas of cancer pain, depression, fatigue and loss of function (see Table 2).

## 4. Conclusions

This intervention project, designed to increase the ability of American Indian cancer survivors to better manage cancer-related symptoms such as pain, depression, fatigue and loss of function, improves communication with health care providers, thereby improving cancer survivors’ quality of life. The Cancer Symptom Management Toolkit was designed specifically for American Indian cancer survivors and their caregivers, and was shaped by participants in earlier phases of the study. The intervention curriculum promoted culturally guided self-management strategies for cancer survivors and, in addition, provided caregivers with tangible ways to offer support to their loved ones. Study outcomes document the intervention was successful in improving knowledge and perceived skills/strategies in the management of all tested domains of cancer-related symptoms.

When examining pre-test scores, the survivors’ responses to pre-test questions present a picture of cancer survivors who had little prior knowledge of cancer-related symptom management, and had limited instruction in pain, depression, fatigue or loss of function management. Survivors reported daily pain and episodes of depression experienced from diagnosis to treatment and beyond. More than one-half to two-thirds of participants suffered from cancer-related pain, yet the majority of survivors who reported that they were prescribed pain medication would take it only during the most painful episodes. They indicated fear of side effects from both of their medications used to treat pain and depression, as well as a fear of addiction to pain medications. Additionally, little was known by participants about pain control and depressive symptoms and the ability to manage or control these symptoms. As to fatigue experiences, most cancer survivors experienced some cancer-related fatigue, however, over one-half had never thought about their ability to manage their fatigue.

Interventions to improve knowledge and cancer symptom management skills and strategies contribute toward improving quality of life during cancer survivorship journeys. Developing relevant, culturally appropriate, effective cancer-related interventions to meet the needs of a diverse set views of health and wellness amongst Indigenous peoples continues to be needed [11,27,28]. A large majority of interventional studies in cancer with Indigenous survivors have shown positive effects on study outcomes, including increasing cancer knowledge, social and spiritual support, cancer service access and communication, yet ours may be the first to demonstrate a change in physical symptom self-management skills [11]. In this study, significant improvement in all targeted domains of cancer symptom management was achieved. This was observed through survivors’ level of knowledge and perceived ability to manage their cancer-related pain, knowledge of depressive symptoms and perceived ability to adopt recommended skills in managing depression, fatigue and function.

The findings from this study guides researchers and healthcare providers towards a better understanding of the meaning and impact of cancer symptoms among American Indian cancer survivors. As others have observed [15,28,29,30], during this project, this study’s findings found that American Indians in the Southwest experienced late diagnosis of their cancer all too often, leading to late-stage cancer at diagnosis and therefore poorer prognoses. In addition, once diagnosed, many cancer survivors lacked effective self-management strategies for the commonly experienced symptoms of pain, fatigue, depression and loss of function. Results from this study demonstrate the impact of the culturally appropriate use of the Talking Circle and toolkit intervention on participants’ knowledge, attitudes, and ability to manage common cancer symptoms. Moreover, participant satisfaction with materials and experience with the Talking Circles curriculum was very high, with multiple indications for the need to expand the project’s reach beyond the study. There are some limitations to this study; for example, the findings of the study may not be generalizable to American Indians living outside of the Southwest, and may not extend to all males since the majority of the study’s participants were female. Participant bias may have affected study data. In addition, the toolkit, which was specifically tailored for Southwestern American Indian cultural preferences, would require adaptation in order to be appropriate for other American Indian communities.

## Figures and Tables

**Figure 1 cancers-14-04771-f001:**
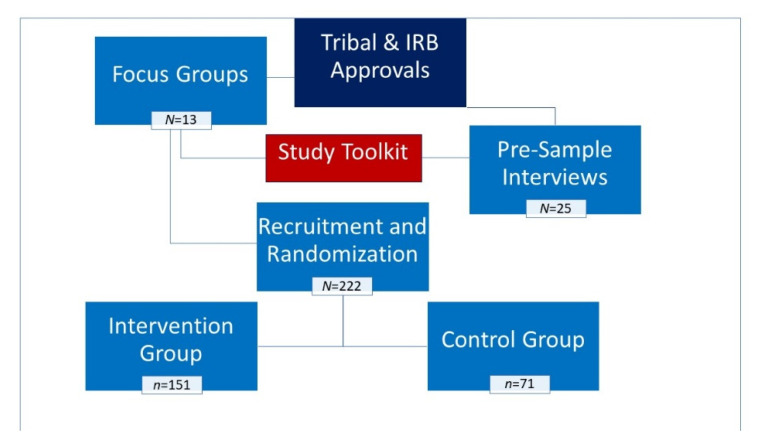
Flow diagram of study design.

**Table 1 cancers-14-04771-t001:** Baseline characteristics (*N* = 222).

Characteristics	Overall (*n* = 222)	Intervention (*n* = 151)	Control (*n* = 71)	*p*-Value
	(Mean/SE)	(Mean/SE)	(Mean/SE)	
Age	42.58 (15.68)	42.12 (15.88)	43.68 (15.28)	0.47
	%	%	%	
Gender				0.11
Male	29.86	26.49	37.14	
Female	70.14	73.51	62.86	
Education				<0.0001
<HS degree	63.06	72.19	43.66	
≥HS degree	36.94	27.81	56.34	
Marital Status				0.01
Currently married	34.93	29.08	47.06	
Divorced/Separated/Widowed/Single	65.07	70.92	52.94	
Employed				0.008
Yes	31.53	25.83	43.66	
No	68.47	74.17	56.34	
Cancer Diagnosis and History				
Type of Cancer (*n* = 50)				0.29
Sarcoma	8.00	96.67	85.00	
Carcinoma	92.00	3.33	15.00	
Currently being treated (*n* = 187)				0.85
Yes	55.61	56.10	54.69	
No	44.39	43.90	45.31	
Type of treatment				
Chemotherapy	30.18	24.50	42.25	0.007
Radiotherapy	13.96	14.57	12.68	0.70
Surgery	11.26	11.26	11.27	1.00
Traditional method	3.15	1.99	5.63	0.21
Other	7.66	8.61	5.63	0.59

**Table 2 cancers-14-04771-t002:** Pre-test–Post-test knowledge changes.

Symptom	Measure	*p*-Value
Pain	At post-test 67.9% of participants reported that they now knew how to manage their pain, a significant increase compared to the control group.	0.003
Depression	At post-test, participants knew how to manage depression as compared to the control group. This was an increase in knowledge levels from 35.1% at pre-test to 66.4% at post-test.	0.004
Fatigue	A statistically significant increase was reported at post-test among those participants who now knew how to manage their fatigue (67.2% post-test vs. 32.5% pre-test). The control group reported no change.	0.0001
Loss of Function	At post-test, a statistically significant increase was reported among participants who now knew how to manage their loss of function (61.9% post-test vs. 28.5% pre-test). The control group reported no increase.	0.0001

## Data Availability

The data presented in this study are available on request from the corresponding author. The data are not publicly available due to confidentiality agreements with Tribes.

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
