# Peer review of "Cancer-Related Symptom Management Intervention for Southwest American Indians"

_cancers, 2022, doi:10.3390/cancers14194771_

Round 1

Reviewer 1 Report

This is an outstanding study and manuscript. It is ready for publication in its current form. The trial was in itself was a challenge to complete and the results are important for future care of Native cancer survivors. Culturally appropriate interventions for American Indian cancer survivors is needed.

Author Response

Thank you for your valuable time and support of this manuscript for publication.

Reviewer 2 Report

The Authors have to be congratulated for conducting such an interesting study. I have no special concerns, and I think that it deserves publication. The style is somehow unusual for scientific literature, but it was a pleasure to read this paper.

Only one minor suggestion, the addition of a picture summarizing the study design would be welcome.

Author Response

The authors wish to thank you for your valuable time and input. 

We have included a figure to illustrate the study design. Please see the Materials & Methods section of the revised manuscript.

Reviewer 3 Report

The study addresses a very important question, and the study design is great.

1. Were any validated questions used to define cancer symptoms and quality of life?

2. Can the authors explain why the men suffered from higher functional status decline? And how was functional status data collected?

Kudos to the authors for designing a great study, and hopefully, it can be scaled up to other populations soon. 

Author Response

Thank you for your careful review of our paper as well as your questions.

Were any validated questions used to define cancer symptoms and quality of life?

RESPONSE: Some validated measures were used to define cancer symptoms and quality of life, such as the CES-D, common pain scales, and components from the SF-36.

Can the authors explain why the men suffered from higher functional status decline? And how was functional status data collected?

RESPONSE:  Directed questions about the experience of limitations due to cancer in activities of daily living such as mobility, work, social events, and self-care were asked.

Sex and gender have historically been foundational to roles, traditions, and ceremonies for many Indigenous Peoples. We suspect men reported higher functional status decline due differing cultural expectations for males. Functional decline reports may also be impacted by cultural differences in self-expression between men and women.